# On Algorithms for Sparse Multi-factor NMF

**Siwei Lyu**    **Xin Wang**

Computer Science Department
University at Albany, SUNY
Albany, NY 12222
{slyu,xwang26}@albany.edu

## Abstract

Nonnegative matrix factorization (NMF) is a popular data analysis method, the objective of which is to approximate a matrix with all nonnegative components into the product of two nonnegative matrices. In this work, we describe a new simple and efficient algorithm for multi-factor nonnegative matrix factorization (mfNMF) problem that generalizes the original NMF problem to more than two factors. Furthermore, we extend the mfNMF algorithm to incorporate a regularizer based on the Dirichlet distribution to encourage the sparsity of the components of the obtained factors. Our sparse mfNMF algorithm affords a closed form and an intuitive interpretation, and is more efficient in comparison with previous works that use fix point iterations. We demonstrate the effectiveness and efficiency of our algorithms on both synthetic and real data sets.

## 1   Introduction

The goal of nonnegative matrix factorization (NMF) is to approximate a nonnegative matrix $V$ with the product of two nonnegative matrices, as $V \approx W_1 W_2$. Since the seminal work of [1] that introduces simple and efficient multiplicative update algorithms for solving the NMF problem, it has become a popular data analysis tool for applications where nonnegativity constraints are natural [2].

In this work, we address the *multi-factor NMF* (mfNMF) problem, where a nonnegative matrix $V$ is approximated with the product of $K \geq 2$ nonnegative matrices, $V \approx \prod_{k=1}^{K} W_k$. It has been argued that using more factors in NMF can improve the algorithm's stability, especially for ill-conditioned and badly scaled data [2]. Introducing multiple factors into the NMF formulation can also find practical applications, for instance, extracting hierarchies of topics representing different levels of abstract concepts in document analysis or image representations [2].

Many practical applications also require the obtained nonnegative factors to be sparse, i.e., having many zero components. Most early works focuses on the matrix $\ell_1$ norm [6], but it has been pointed out that $\ell_1$ norm becomes completely toothless for factors that have constant $\ell_1$ norms, as in the case of stochastic matrices [7, 8]. Regularizers based on the entropic prior [7] or the Dirichlet prior [8] have been shown to be more effective but do not afford closed-form solutions.

The main contribution of this work is therefore two-fold. First, we describe a new algorithm for the mfNMF problem. Our solution to mfNMF seeks optimal factors that minimize the total difference between $V$ and $\prod_{k=1}^{K} W_k$, and it is based on the solution of a special matrix optimization problem that we term as the *stochastic matrix sandwich* (SMS) problem. We show that the SMS problem affords a simple and efficient algorithm consisting of only multiplicative update and normalization (Lemma 2). The second contribution of this work is a new algorithm to incorporate the Dirichlet sparsity regularizer in mfNMF. Our formulation of the sparse mfNMF problem leads to a new closed-form solution, and the resulting algorithm naturally embeds sparsity control into the mfNMF algorithm without iteration (Lemma 3). We further show that the update steps of our sparse

mfNMF algorithm afford a simple and intuitive interpretation. We demonstrate the effectiveness and efficiency of our sparse mfNMF algorithm on both synthetic and real data.

## 2 Related Works

Most exiting works generalizing the original NMF problem to more than two factors are based on the multi-layer NMF formulation, in which we solve a sequence of two factor NMF problems, as $V \approx W_1 H_1, H_1 \approx W_2 H_2, \cdots$, and $H_{K-1} \approx W_{K-1} W_K$ [3–5]. Though simple and efficient, such greedy algorithms are not associated with a consistent objective function involving all factors simultaneously. Because of this, the obtained factors may be suboptimal measured by the difference between the target matrix $V$ and their product. On the other hand, there exist much fewer works directly solving the mfNMF problem, one example is a multiplicative algorithm based on the general Bregmann divergence [9]. In this work, we focus on the generalized Kulback-Leibler divergence (a special case of the Bregmann divergence), and use its decomposing property to simplify the mfNMF objective and remove the undesirable multitude of equivalent solutions in the general formulation. These changes lead to a more efficient algorithm that usually converges to a better local minimum of the objective function in comparison of the work in [9] (see Section 6).

As a common means to encouraging sparsity in machine learning, the $\ell_1$ norm has been incorporated into the NMF objective function [6] as a sparsity regularizer. However, the $\ell_1$ norm may be less effective for nonnegative matrices, for which it reduces to the sum of all elements and can be decreased trivially by scaling down all factors without affecting the number of zero components. Furthermore, the $\ell_1$ norm becomes completely toothless in cases when the nonnegative factors are constrained to have constant column or row sums (as in the case of stochastic matrices).

An alternative solution is to use the Shannon entropy of each column of the matrix factor as sparsity regularizer [7], since a vector with unit $\ell_1$ norm and low entropy implies that only a few of its components are significant. However, the entropic prior based regularizer does not afford a closed form solution, and an iterative fixed point algorithm is described based on the Lamber's W-function in [7]. Another regularizer is based on the symmetric Dirichlet distribution with concentration parameter $\alpha < 1$, as such a model allocates more probability weights to sparse vectors on a probability simplex [8, 10, 11]. However, using the Dirichlet regularizer has a practical problem, as it can become unbounded when there is a zero element in the factor. Simply ignoring such cases as in [11] can lead to unstable algorithm (see Section 5.2). Two approaches have been described to solve this problem, one is based on the constrained concave-convex procedure (CCCP) [10]. The other uses the psuedo-Dirichlet regularizer [8], which is a bounded perturbation of the original Dirichlet model. A drawback common to these methods is that they require extra iterations for the fix point algorithm. Furthermore, the effects of the updating steps on the sparsity of the resulting factors are obscured by the iterative steps. In contrast, our sparse mfNMF algorithm uses the original Dirichlet model and does not require extra fix point iteration. More interestingly, the update steps of our sparse mfNMF algorithm afford a simple and intuitive interpretation.

## 3 Basic Definitions

We denote $\mathbf{1}_m$ as the all-one column vector of dimension $m$ and $I_m$ as the $m$-dimensional identity matrix, and use $V \geq 0$ or $\mathbf{v} \geq 0$ to indicate that all elements of a matrix $V$ or a vector $\mathbf{v}$ are nonnegative. Throughout the paper, we assume a matrix does not have all-zero columns or rows. An $m \times n$ nonnegative matrix $V$ is *stochastic* if $V^T \mathbf{1}_m = \mathbf{1}_n$, i.e., each column has a total sum of one. Also, for stochastic matrices $W_1$ and $W_2$, their product $W_1 W_2$ is also stochastic. Furthermore, an $m \times n$ nonnegative matrix $V$ can be uniquely represented as $V = SD$, with an $n \times n$ nonnegative diagonal matrix $D = \text{diag}(V^T \mathbf{1}_m)$ and an $m \times n$ stochastic matrix $S = V D^{-1}$.

For nonnegative matrices $V$ and $W$, their generalized Kulback-Leibler (KL) divergence [1] is defined as

$$d(V, W) = \sum_{i=1}^{m} \sum_{j=1}^{n} \left( V_{ij} \log \frac{V_{ij}}{W_{ij}} - V_{ij} + W_{ij} \right). \tag{1}$$

We have $d(V,W) \geq 0$ and the equality holds if and only if $V = W$[1]. We emphasize the following decomposition of the generalized KL divergence: representing $V$ and $W$ as products of stochastic matrices and diagonal matrices, $V = S^{(V)}D^{(V)}$ and $W = S^{(W)}D^{(W)}$, we can decompose $d(V,W)$ into two terms involving only stochastic matrices or diagonal matrices, as

$$d(V,W) = d\left(V, S^{(W)}D^{(V)}\right) + d\left(D^{(V)}, D^{(W)}\right) = \sum_{i=1}^{m}\sum_{j=1}^{n} V_{ij} \log\left[\frac{S_{ij}^{(V)}}{S_{ij}^{(W)}}\right] + d\left(D^{(V)}, D^{(W)}\right).$$

(2)

Due to space limit, the proof of Eq.(2) is deferred to the supplementary materials.

## 4 Multi-Factor NMF

In this work, we study the multi-factor NMF problem based on the generalized KL divergence. Specifically, given an $m \times n$ nonnegative matrix $V$, we seek $K \geq 2$ matrices $W_k$ of dimensions $l_{k-1} \times l_k$ for $k = 1, \cdots, K$, with $l_0 = m$ and $l_K = n$ that minimize $d(V, \prod_{k=1}^{K} W_k)$, s.t., $W_k \geq 0$.

This simple formulation has a drawback as it is invariant to relative scalings between the factors: for any $\gamma > 0$, we have $d(V, W_1 \cdots W_i \cdots W_j \cdots W_K) = d(V, W_1 \cdots (\gamma W_i) \cdots (\frac{1}{\gamma}W_j) \cdots W_K)$. In other words, there exist infinite number of equivalent solutions, which gives rise to an intrinsic ill-posed nature of the mfNMF problem.

To alleviate this problem, we constrain the first $K-1$ factors, $W_1, \cdots, W_{K-1}$, to be stochastic matrices, and differentiate the notationns with $X_1, \cdots, X_{K-1}$. Using the property of nonnegative matrices, we represent the last nonnegative factor $W_K$ as the product of a stochastic matrix $X_K$ and a nonnegative diagonal matrix $D^{(W)}$. As such, we represent the nonnegative matrix $\prod_{k=1}^{K} W_k$ as the product of a stochastic matrix $S^{(W)} = \prod_{k=1}^{K} X_k$ and a diagonal matrix $D^{(W)}$. Similarly, we also decompose the target nonnegative matrix $V$ as the product of a stochastic matrix $S^{(V)}$ and a nonnegative diagonal matrix $D^{(V)}$. It is not difficult to see that any solution from this more constrained formulation leads to a solution to the original problem and vice versa.

Applying the decomposition in Eq.(2), the mfNMF optimization problem can be re-expressed as

$$\min_{X_1,\cdots,X_K,D^{(W)}} \quad d\left(V, \left(\prod_{k=1}^{K} X_k\right)D^{(V)}\right) + d\left(D^{(V)}, D^{(W)}\right)$$

$$\text{s.t.} \quad X_k^T \mathbf{1}_{l_{k-1}} = \mathbf{1}_{l_k}, X_k \geq 0, k = 1, \cdots, K, D^{(W)} \geq 0.$$

As such, the original problem is solved with two sub-problems, each for a different set of variables.

The first sub-problem solves for the diagonal matrix $D^{(W)}$, as:

$$\min_{D^{(W)}} d\left(D^{(V)}, D^{(W)}\right), \text{ s.t. } D^{(W)} \geq 0.$$

Per the property of the generalized KL divergence, its solution is trivially given by $D^{(W)} = D^{(V)}$.

The second sub-problem optimizes the $K$ stochastic factors, $X_1, \cdots, X_K$, which, after dropping irrelevant constants and rearranging terms, becomes

$$\max_{X_1,\cdots,X_K} \sum_{i=1}^{m}\sum_{j=1}^{n} V_{ij} \log\left(\prod_{k=1}^{K} X_k\right)_{ij}, \text{ s.t. } X_k^T \mathbf{1}_{l_{k-1}} = \mathbf{1}_{l_k}, X_k \geq 0, k = 1, \cdots, K. \quad (3)$$

Note that Eq.(3) is essentially the maximization of the similarity between the stochastic part of $V$, $S^V$ with the stochastic matrix formed as the product of the $K$ stochastic matrices $X_1, \cdots, X_K$, weighted by $D^V$.

### 4.1 Stochastic Matrix Sandwich Problem

Before describing the algorithm solving (3), we first derive the solution to another related problem that we term as the *stochastic matrix sandwich* (SMS) problem, from which we can construct a solution to (3). Specifically, in an SMS problem one minimizes the following objective function with regards to an $m' \times n'$ stochastic matrix $X$, as

$$\max_X \sum_{i=1}^{m} \sum_{j=1}^{n} C_{ij} \log\left(AXB\right)_{ij}, \ \text{s.t.} \ X^T \mathbf{1}_{m'} = \mathbf{1}_{n'}, X \geq 0, \tag{4}$$

where $A$ and $B$ are two known stochastic matrices of dimensions $m \times m'$ and $n' \times n$, respectively, and $C$ is an $m \times n$ nonnegative matrix.

We note that (4) is a convex optimization problem [12], which can be solved with general numerical procedures such as interior-point methods. However, we present here a new simple solution based on multiplicative updates and normalization, which completely obviates control parameters such as the step sizes. We first show that there exists an "auxiliary function" to $\log\left(AXB\right)_{ij}$.

**Lemma 1.** *Let us define*

$$\mathcal{F}_{ij}(X; \tilde{X}) = \sum_{k=1}^{m'} \sum_{l=1}^{n'} \frac{A_{ik}\tilde{X}_{kl}B_{lj}}{\left(A\tilde{X}B\right)_{ij}} \log\left(\frac{X_{kl}}{\tilde{X}_{kl}}\left(A\tilde{X}B\right)_{ij}\right),$$

*then we have* $\mathcal{F}_{ij}(X; \tilde{X}) \leq \log\left(AXB\right)_{ij}$ *and* $\mathcal{F}_{ij}(X; X) = \log\left(AXB\right)_{ij}$.

Proof of Lemma 1 can be found in the supplementary materials.

Based Lemma 1 we can develop an EM style iterative algorithm to optimize (4), in which, starting with an initial values $X = X_0$, we iteratively solve the following optimization problem,

$$X_{t+1} \leftarrow \operatorname*{argmax}_{X:X^T\mathbf{1}_{m'}=\mathbf{1}_{n'}, X\geq 0} \sum_{i=1}^{m} \sum_{j=1}^{n} C_{ij}\mathcal{F}_{ij}(X; X_t) \ \text{and} \ t \leftarrow t+1. \tag{5}$$

Using the relations given in Lemma 1, we have:

$$\sum_{i,j} C_{ij} \log\left(AX_tB\right)_{ij} = \sum_{i,j} C_{ij}\mathcal{F}_{ij}(X_t; X_t) \leq \sum_{i,j} C_{ij}\mathcal{F}_{ij}(X_{t+1}; X_t) \leq \sum_{i,j} C_{ij} \log\left(AX_{t+1}B\right)_{ij},$$

which shows that each iteration of (5) leads to feasible $X$ and does not decrease the objective function of (4). Rearranging terms and expressing results using matrix operations, we can simplify the objective function of (5) as

$$\sum_{i,j} C_{ij}\mathcal{F}_{ij}(X; \tilde{X}) = \sum_{k=1}^{m'} \sum_{l=1}^{n'} M_{kl} \log X_{kl} + \text{const}, \tag{6}$$

where

$$M = \tilde{X} \otimes \left[A^T\left(C \oslash \left(A\tilde{X}B\right)\right)B^T\right], \tag{7}$$

where $\otimes$ and $\oslash$ correspond to element-wise matrix multiplication and division, respectively. A detailed derivation of (6) and (7) is given in the supplemental materials. The following result shows that the resulting optimization has a closed-form solution.

**Lemma 2.** *The global optimal solution to the following optimization problem,*

$$\max_X \sum_{k=1}^{m'} \sum_{l=1}^{n'} M_{kl} \log X_{kl}, \ \text{s.t.} \ X^T\mathbf{1}_{m'} = \mathbf{1}_{n'}, X \geq 0, \tag{8}$$

*is given by*

$$X_{kl} = \frac{M_{kl}}{\sum_k M_{kl}}.$$

Proof of Lemma 2 can be found in the supplementary materials.

Next, we can construct a coordinate-wise optimization solution to the mfNMF problem (3) that iteratively optimizes each $X_k$ while fixing the others, based on the solution to the SMS problem given in Lemma 2. In particular, it is easy to see that for $C = V$,

- solving for $X_1$ with fixed $X_2, \cdots, X_K$ is an SMS problem with $A = I_m$, $X = X_1$ and $B = \prod_{k=2}^{K} X_k$;
- solving for $X_k$, $k = 2, \cdots, K-1$, with fixed $X_1, \cdots, X_{k-1}, X_{k+1}, \cdots, X_K$ is an SMS with $A = \prod_{k'=1}^{k-1} X_{k'}$, $X = X_k$, and $B = \prod_{k'=k+1}^{K} X_{k'}$;
- and solving for $X_K$ with fixed $X_1, \cdots, X_{K-1}$ is an SMS problem with $A = \prod_{k=1}^{K-1} X_k$, $X = X_K$ and $B = I_n$.

In practice, we do not need to run each SMS optimization to converge, and the algorithm can be implemented with a few fixed steps updating each factor in order.

It should be pointed out that even though SMS is a convex optimization problem guaranteed with a global optimal solution, this is not the case for the general mfNMF problem (3), the objective function of which is non-convex (it is an example of the multi-convex function [13]).

# 5   Sparse Multi-Factor NMF

Next, we describe incorporating sparsity regularization in the mfNMF formulation. We assume that the sparsity requirement is applied to each individual factor in the mfNMF objective function (3), as

$$\max_{X_1,\cdots,X_K} \sum_{i,j} V_{ij} \log \left( \prod_{k=1}^{K} X_k \right)_{ij} + \sum_{k=1}^{K} \ell(X_k), \ \ \text{s.t.} \ X_k^T \mathbf{1}_{l_{k-1}} = \mathbf{1}_{l_k}, X_k \geq 0, \tag{9}$$

where $\ell(X)$ is the sparsity regularizer that is larger for stochastic matrix $X$ with more zero entries. As the overall mfNMF can be solved by optimizing each individual factor in an SMS problem, we focus here on the case where the sparsity regularizer of each factor is introduced in (4), to solve

$$\max_X \sum_{i,j} C_{ij} \log (AXB)_{ij} + \ell(X), \ \ \text{s.t.} \ X^T \mathbf{1}_{m'} = \mathbf{1}_{n'}, X \geq 0. \tag{10}$$

## 5.1   Dirichlet Sparsity Regularizer

As we have mentioned, the typical choice of $\ell(\cdot)$ as the matrix $\ell_1$ norm is problematic in (10), as $\|X\|_1 = \sum_{ij} X_{ij} = n'$ is a constant. On the other hand, if we treat each column of $X$ as a point on a probability simplex, as their elements are nonnegative and sum to one, then we can induce a sparse regularizer from the Dirichlet distribution. Specifically, a Dirichlet distribution of $d$-dimensional vectors $\mathbf{v} : \mathbf{v} \geq 0, \mathbf{1}^T \mathbf{v} = 1$ is defined as $\mathcal{D}\text{ir}(\mathbf{v}; \alpha) = \frac{\Gamma(d\alpha)}{\Gamma(\alpha)^d} \prod_{k=1}^{d} v_k^{\alpha-1}$, where $\Gamma(\cdot)$ is the standard Gamma function[2]. The parameter $\alpha \in [0,1]$ is the parameter that controls the sparsity of samples – smaller $\alpha$ corresponds to higher likelihood of sparse $\mathbf{v}$ in $\mathcal{D}\text{ir}(\mathbf{v}; \alpha)$.

Incorporating a Dirichlet regularizer with parameter $\alpha_l$ to each column of $X$ and dropping irrelevant constant terms, (10) reduces to[3]

$$\max_X \sum_{i=1}^{m} \sum_{j=1}^{n} C_{ij} \log (AXB)_{ij} + \sum_{k=1}^{m'} \sum_{l=1}^{n'} (\alpha_l - 1) \log X_{kl}, \ \ \text{s.t.} \ X^T \mathbf{1}_{m'} = \mathbf{1}_{n'}, X \geq 0. \tag{11}$$

As in the case of mfNMF, we introduce the auxiliary function of $\log(AXB)_{ij}$ to form an upper-bound of (11) and use an EM-style algorithm to optimize (11). Using the result given in Eqs.(6) and (7), the optimization problem can be further simplified as:

$$\max_X \sum_{k=1}^{m'} \sum_{l=1}^{n'} (M_{kl} + \alpha_l - 1) \log X_{kl}, \ \ \text{s.t.} \ X^T \mathbf{1}_{m'} = \mathbf{1}_{n'}, X \geq 0. \tag{12}$$

## 5.2   Solution to SMS with Dirichlet Sparse Regularizer

However, a direct optimization of (12) is problematic when $\alpha_l < 1$: if there exists $M_{kl} < 1 - \alpha_l$, the objective function of (12) becomes non-convex and unbounded – the term $(M_{kl} + \alpha_l - 1) \log X_{kl}$ approaches $\infty$ as $X_{kl} \to 0$. This problem is addressed in [8] by modifying the definition of the Dirichlet regularizer in (11) to $(\alpha_l - 1) \log(X_{kl} + \epsilon)$, where $\epsilon > 0$ is a predefined parameter. This avoids the problem of taking logarithm of zero, but it leads to a less efficient algorithm based on an iterative fix point procedure. In addition, the fix point algorithm is difficult to interpret as its effect on the sparsity of the obtained factors is obscured by the iterative steps.

On the other hand, notice that if we tighten the nonnegativity constraint to $X_{kl} \geq \epsilon$, the objective function of (12) will always be finite. Therefore, we can simply modify the optimization of (12) the objective function to become infinity, as:

$$\max_X \sum_{k=1}^{m'} \sum_{l=1}^{n'} (M_{kl} + \alpha_l - 1) \log X_{kl}, \ \ \text{s.t.} \ X^T \mathbf{1}_{m'} = \mathbf{1}_{n'}, X_{kl} \geq \epsilon, \forall k, l. \tag{13}$$

The following result shows that with a sufficiently small $\epsilon$, the constrained optimization problem in (13) has a unique global optimal solution that affords a closed-form and intuitive interpretation.

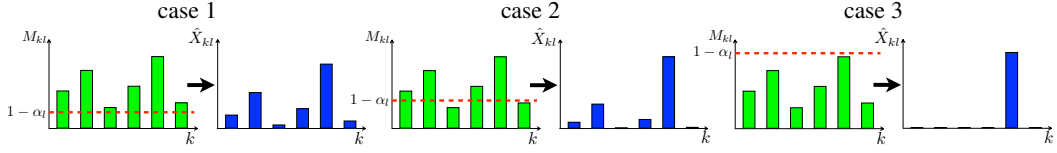

**Figure 1:** *Sparsification effects on the updated vectors before (**left**) and after (**right**) applying the algorithm given in Lemma 3, with each column illustrating one of the three cases.*

**Lemma 3.** *Without loss of generality, we assume $M_{kl} \neq 1 - \alpha_l, \forall k, l$[4]. If we choose a constant $\epsilon \in \left(0, \frac{\min_{kl}\{|M_{kl}+\alpha_l-1|\}}{m' \max_{kl}\{|M_{kl}+\alpha_l-1|\}}\right)$, and for each column $l$ define $\mathcal{N}_l = \{k | M_{kl} < 1 - \alpha_l\}$ as the set of elements with $M_{kl} + \alpha_l - 1 < 0$, then the following is the global optimal solution to (13):*

- **case 1**. $|\mathcal{N}_l| = 0$, *i.e., all constant coefficients of (13) are positive,*

$$\hat{X}_{kl} = \frac{M_{kl} + \alpha_l - 1}{\sum_{k'}[M_{k'l} + \alpha_l - 1]}, \tag{14}$$

- **case 2**. $0 < |\mathcal{N}_l| < m'$, *i.e., the constant coefficients of (13) have mixed signs,*

$$\hat{X}_{kl} = \epsilon \cdot \delta[k \in \mathcal{N}_l] + \frac{(1 - |\mathcal{N}_l|\epsilon)[M_{kl} + \alpha_l - 1]}{\sum_{k'}\{[M_{k'l} + \alpha_l - 1] \cdot \delta[k' \notin \mathcal{N}_l]\}} \cdot \delta[k \notin \mathcal{N}_l], \tag{15}$$

  *where $\delta(c)$ is the Kronecker function that takes 1 if c is true and 0 otherwise.*

- **case 3**. $|\mathcal{N}_l| = m'$, *i.e., all constant coefficients of (13) are negative,*

$$\hat{X}_{kl} = (1 - (m'-1)\epsilon) \cdot \delta\left[k = \operatorname*{argmax}_{k' \in \{1, \cdots, m'\}} M_{k'l}\right] + \epsilon \cdot \delta\left[k \neq \operatorname*{argmax}_{k' \in \{1, \cdots, m'\}} M_{k'l}\right]. \tag{16}$$

Proof of Lemma 3 can be found in the supplementary materials. Note that the algorithm provided in Lemma 3 is still valid when $\epsilon = 0$, but the theoretical result of it attaining the global optimum of a finite optimization problem only holds for $\epsilon$ satisfying the condition in Lemma 3.

We can provide an intuitive interpretation to Lemma 3, which is schematically illustrated in Fig.1 for a toy example. For the first case (first column of Fig.1) when all constant coefficients of (13) are positive, it simply reduces to first decrease every $M_{kl}$ by $1-\alpha_l$ and then renormalize each column to sum to one, Eq.(14). This operation of reducing the same amount from all elements in one column of $M$ has the effect of making "the rich get richer and the poor get poorer" (known as Dalton's 3rd law), which increases the imbalance of the elements and improves the chances of small elements to be reduced to zero in the subsequent steps [15][5]. In the second case (second column of Fig.1), when the coefficients of (13) have mixed signs, the effect of the updating step in (15) is two-fold. For $M_{kl} < 1 - \alpha_l$ (first term in Eq.(15)), they are all reduced to $\epsilon$, which is the *de facto* zero. In other words, components below the threshold $1 - \alpha_l$ are eliminated to zero. On the other hands, terms with $M_{kl} > 1 - \alpha_l$ (second term in Eq.(15)) are redistribute with the operation of reduction of $1 - \alpha_l$ followed by renormalization. In the last case when all coefficients of (13) are negative (third column of Fig.1), only the element corresponding to $M_{kl}$ that is closest to the threshold $1 - \alpha_k$, or equivalently, the largest of all $M_{kl}$, will survive with a non-zero value that is essentially 1 (first term in Eq.(16)), while the rest of the elements all become extinct (second term in Eq.(16)), analogous to a scenario of "survival of the fittest". Note that it is the last two cases actually generating zero entries in the factors, but the first case makes more entries suitable for being set to zero. The thresholding and renormalization steps resemble algorithms in sparse coding [16].

## 6 Experimental Evaluations

We perform experimental evaluations of the sparse multi-factor NMF algorithm using synthetic and real data sets. In the first set of experiments, we study empirically the convergence of the multiplicative algorithm for the SMS problem (Lemma 2). Specifically, with several different choices of

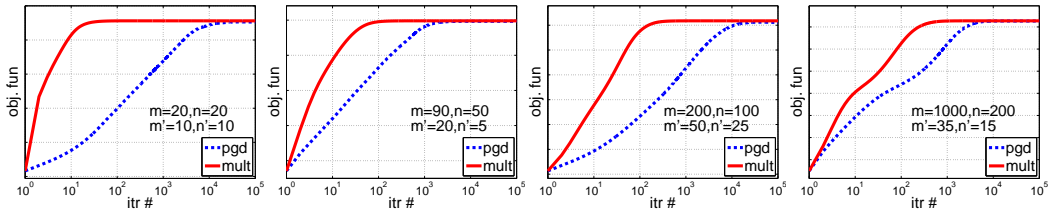

**Figure 2:** *Convergences of the SMS objective function with multiplicative update algorithm (*`mult`* solid curve) and the projected gradient ascent method (*`pgd`* dashed curve) for different problem sizes.*

$(m, n, m', n')$, we randomly generate stochastic matrices $A$ $(m \times m')$ and $B$ $(n' \times n)$, and non-negative matrix $C$ $(m \times n)$. We then apply the SMS algorithm to solve for the optimal $X$. We compare our algorithm with a projected gradient ascent optimization of the SMS problem, which updates $X$ using the gradient of the SMS objective function and chooses a step size to satisfy the nonnegative and normalization constraints. We do not consider methods that use the Hessian matrix of the objective function, as constructing a general Hessian matrix in this case have prohibitive memory requirement even for a medium sized problem. Shown in Fig.2 are several runs of the two algorithms starting at the same initial values, as the the objective function of SMS vs. the number of updates of $X$. Because of the convex nature of the SMS problem, both algorithms converge to the same optimal value regardless of the initial values. On the other hand, the multiplicative updates for SMS usually achieve two order speed up in terms of the number of iterations and typically about 10 times faster in running time when compared to the gradient based algorithm.

In the second set of experiments, we evaluate the performance of the coordinate-update mfNMF algorithm based on the multiplicative updating algorithm of the SMS problem (Section 4.1). Specifically, we consider the mfNMF problem that approximates a randomly generated target nonnegative matrix $V$ of dimension $m \times n$ with the product of three stochastic factors, $W_1$ $(m \times m')$, $W_2$ $(m' \times n')$, and $W_3$ $(n' \times n)$. The performance of the algorithm is evaluated by the logarithm of the generalized KL divergence for between $V$ and $W_1 W_2 W_3$, of which lower numerical values suggest better performances. As a comparison, we also implemented a multi-layer NMF algorithm [5], which solves two NMF problems in sequence, as: $V \approx W_1 \tilde{V}$ and $\tilde{V} \approx W_2 W_3$, and the multiplicative update algorithm of mfNMF of [9], both of which are based on the generalized KL divergence. To make the comparison fair, we start all three algorithms with the same initial values.

| | $m, n, m', n'$ | | | |
|---|---|---|---|---|
| | 50,40,30,10 | 200,100,60,30 | 1000,400,200,50 | 5000,2000,100,20 |
| multi-layer NMF [5] | 1.733 | 2.595 | 70.526 | 183.617 |
| multi-factor NMF [9] | 1.431 | 2.478 | 66.614 | 174.291 |
| multi-factor NMF (this work) | **1.325** | **2.340** | **62.086** | **161.338** |

**Table 1:** *Comparison of the multi-layer NMF method and two mfNMF methods for a three factor with different problem sizes. The values correspond to the logarithm of generalized KL divergence, $\log d(V, W_1 W_2 W_3)$. Lower numerical values (in bold font) indicate better performances.*

The results of several runs of these algorithms for different problem sizes are summarized in Table 1, which show that in general, mfNMF algorithms lead to better solutions corresponding to lower generalized KL divergences between the target matrix and the product of the three estimated factors. This is likely due to the fact that these algorithms optimize the generalized KL divergence directly, while multi-layer NMF is a greedy algorithm with sub-optimal solutions. On the other hand, our mfNMF algorithm consistently outperforms the method of [9] by a significant margin, with on average 40% less iterations. We think the improved performance and running efficiency are due to our formulation of the mfNMF problem based on stochastic matrices, which reduces the solution space and encourage convergence to a better local minimum of the objective function.

We apply the sparse mfNMF algorithm to data converted from grayscale images from the *MNIST Handwritten Digits* data set [17] that are vectorized to column vectors and normalized to have total sum of one. All vectorized and normalized images are collected to form the target stochastic matrix $V$, which are decomposed into the product of three factors $W_1 W_2 W_3$. We also incorporate the Dirichlet sparsity regularizers with different configurations. For simplicity, we use the same parameter for all column vectors in one factor. The threshold is set as $\epsilon = 10^{-8}/n$ where $n$ is the total number of images. Shown in Fig.3 are the decomposition results corresponding to 500 vectorized $20 \times 20$ images of handwritten digit 3, that are decomposed into three factors of size $400 \times 196$,

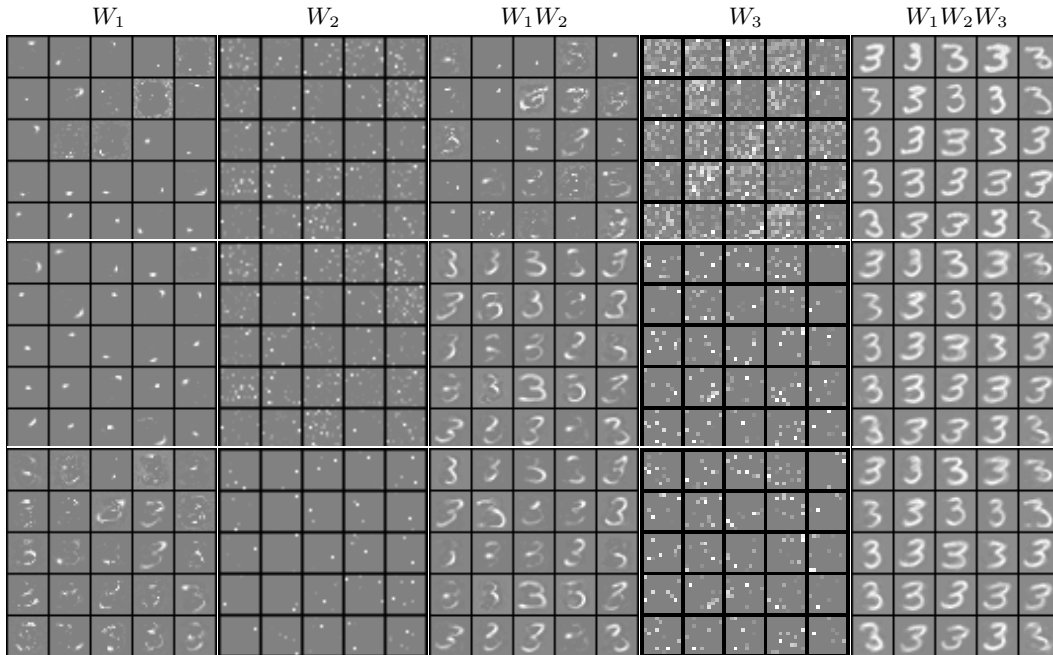

**Figure 3:** *Sparse mfNMF algorithm on the handwritten digit images. The three rows correspond to three cases as: $\alpha_1 = 1, \alpha_2 = 1, \alpha_3 = 1$, $\alpha_1 = 1, \alpha_2 = 1, \alpha_3 = 0.99$, $\alpha_1 = 1, \alpha_2 = 0.99, \alpha_3 = 0.99$, respectively. See texts for more details.*

$196 \times 100$, and $100 \times 500$. The columns of the factors are reshaped to shown as images, where the brightness of each pixel in the figure is proportional to the nonnegative values in the corresponding factors. Due to space limit, we only show the first 25 columns in each factor. All three factorization results can reconstruct the target matrix (last column), but they put different constraints on the obtained factors. The factors are also visually meaningful: factor $W_1$ contains low level components of the images that when combined with factor $W_2$ forms more complex structures. The first row corresponds to running the mfNMF without sparsity regularizer. The two rows below correspond to the cases when the Dirichlet sparsity regularizer is applied to the third factor and to the second and third factor simultaneously. Compare with the corresponding results in the non-sparse case, the obtained factors contain more zeros. As a comparison, we also implement mfNMF algorithm using a pseudo-Dirichlet sparse regularizer [8]. With similar decomposition results, our algorithm is typically $3 - 5$ times faster as it does not require the extra iterations of a fix point algorithm.

## 7 Conclusion

We describe in this work a simple and efficient algorithm for the sparse multi-factor nonnegative matrix factorization (mfNMF) problem, involving only multiplicative update and normalization. Our solution to incorporate Dirichlet sparse regularizer leads to a closed form solution and the resulting algorithm is more efficient than previous works based on fix point iterations. The effectiveness and efficiency of our algorithms are demonstrated on both synthetic and real data sets.

There are several directions we would like to further explore. First, we are studying if similar multiplicative update algorithm also exists for mfNMF with more general similarity norms such as Csizar's divergence [18], Itakura-Saito divergence, [19], $\alpha$-$\beta$ divergence [20] or the Bregmann divergence [9]. We will also study incorporating other constraints (e.g., value ranges) over the factors into the mfNMF algorithm. Last, we would like to further study applications of mfNMF in problems such as co-clustering or hierarchical document topic analysis, exploiting its ability to recover hierarchical decomposition of nonnegative matrices.

## Acknowledgement

This work is supported by the National Science Foundation under Grant Nos. IIS-0953373, IIS-1208463 and CCF-1319800.

## Footnotes

[1]In computing the generalized KL divergence, we define $0 \log 0 = 0$ and $\frac{0}{0} = 0$.

[2]For simplicity, we only discuss the symmetric Dirichlet model, but the method can be easily extended to the non-symmetric Dirichlet model with different $\alpha$ value for different dimension.

[3]Alternatively, this special case of NMF can be formulated as $C = AXB + E$, where $E$ contains independent Poisson samples [14], and (11) can be viewed as a (log) *maximum a posteriori* estimation of column vectors of $X$ with a Poisson likelihood and symmetric Dirichlet prior.

[4]It is easy to show that the optimal solution in this case is $X_{kl} = 0$, i.e., setting the corresponding component in $X$ to zero. So we can technically ignore such elements for each column index $l$.

[5]Some early works (e.g., [11]) obtain simpler solution by setting negative $M_{k'l} + \alpha_l - 1$ to zero followed by normalization. Our result shows that such a solution is not optimal.

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
