[Reviews · NeurIPS 2013]

Submitted by Assigned_Reviewer_5

The paper proposes a new multiplicative algorithm for multi-factor nonnegative matrix
factorization (mfNMF). The closest previous research is [3] (in Section 4.4),
where the target matrix is approximated based on the general Bregman divergence
minimized by a multiplicative algorithm.
The current paper proposes an efficient algorithm specialized for the general
Kullback-Leibler (KL) divergence, a special case of Bregman divergence.
Experiments show that the proposed algorithm outperforms the algorithm proposed
in [3] for the general KL divergence, as well as another type of mfNMF called
multilayer NMF [7-9].
The paper also proposes a sparse variant of mfNMF with a Dirichlet regularizer.
In the proposed framework, the Dirichlet regularizer behaves like a conjugate prior,
and the sparse variant is built with a simple modification (Eq.(8) to Eq.(12))
of the objective.
The obtained algorithm (Lemma 4) seems like ones for Bayesian automatic relevance determination.

I think this is an interesting paper. The main idea for deriving a new algorithm
is to decompose the target matrix into the product of a stochastic matrix and a
diagonal scale matrix, and then simplify the objective based on the decomposition (2)
of the generalized KL divergence.
This alleviates the ill-posedness of mfNMF, and results in better performance
(better quality of the found local minima), and efficient computation, as shown in
the experiments. The authors also elaborate on making the algorithm fully
multiplicative by introducing an auxiliary function (Lemma 2) for the partial
problem (4).
The obtained algorithm seems simple to implement, and work fine in the experiments.

Pros:
- The proposed algorithm is simple and experimentally shown to outperform
the previous methods.
- Unique ideas are found: focusing on stochastic matrices, and using
decomposition (2) of the generalized KL divergence. These techniques
might be used for other problems.

Cons:
- Experiments on the MNIST data set do not appeal the usefulness of mfNMF
and sparse mfNMF. I do not understand what mfNMF found in Fig.3
other than what two-factor NMF can find. In other words,
why do you want to decompose W2W3 into W2 and W3?

Quality:
The paper is technically sound, and the claims are supported by the experiments.

Clarity:
The paper is clearly written. But I would suggest the authors to focus on
the closest research [3] in Section 2, and make readers understand
what the differences between [3] and the current paper without need of
checking the reference.

Originality:
The idea to focus on the stochastic matrices (in a problem where
the purpose is not to find stochastic matrices) seems new.

Significance:
Since the proposed algorithm is simple and experimentally shown to outperform
the previous methods, it would be used by practitioners.
The idea of focusing on the stochastic part of the target matrix
might be used by other researchers when they develop new algorithms.

Minor comments:
- In Lemma 1, S = D^-1 V should be S = V D^-1. Lemma 1 seems trivial because
its just a column-wise normalization, so just stating that 'we decompose V as
V = SD' might be ok.
- I suppose that Eq.(3) is the maximization of the similarity between the stochastic
matrices, S^{V} and S^{W}, weighted by D^{V}. If so, noting this would make it clear
that you are focusing on approximating the stochastic part.

Summary: A paper proposing a new algorithm for multi-factor NMF,
which outperforms the previous methods in terms of computation and approximation quality.
The techniques used here might be used for other problems.


Submitted by Assigned_Reviewer_6

The authors present a method for learning sparse multi-factor NMFs globally. The factors are optimized in coordinate-wise fashion using a closed-form solution to the "stochastic matrix sandwich" problem, which I found interesting; furthermore, the multi-factor NMF empirically seems to outperform the previous multi-layer NMF methods.

I have several quibbles with this paper:

1) The motivation for learning multi-factor NMFs in the first place is unclear; e.g., what's the use of decomposing the MNIST data into 3 factors rather than just using standard NMF? The authors should provide the justification at the very beginning in the introduction. I did see a potential motivation in the last sentence of the paper ("hierarchical document topic analysis") but am not sure if there are any other uses of multi-factor NMFs.

2) Is Equation (3) missing a term "- (\prod_k X_k * D^v)_ij" ? Looking at the generalized KL divergence in Eq (1) and the expression in line 136, it seems that the additional term should be there, unless I am missing something? I would appreciate the authors' clarification on this issue.

3) Regarding the closed-form SMS solution, if the number of matrices K=2 (standard NMF), does the resulting algorithm resemble any other well-known update for NMF, or would it be a novel algorithm as well?

4) It would be very nice if the paper could somehow link the multi-factor NMF with Dirichlet priors to hierarchical LDA models, or at least mention related works (e.g., having topics of topics, etc).

5) It would be nice if Table 1 could have some error bars (maybe achieved through different initializations). Also, how does adjusting the sparsity help for the MNIST data?
Summary: The authors present a method for learning sparse multi-factor NMFs globally. The factors are optimized in coordinate-wise fashion using a closed-form solution to the "stochastic matrix sandwich" problem, which I found interesting; furthermore, the multi-factor NMF empirically seems to outperform the previous multi-layer NMF methods.

Submitted by Assigned_Reviewer_7

This paper describes a new algorithm for the sparse multi-factor
nonnegative matrix factorization (mfNMF) problem, which generalizes
the original NMF problem to more than two factors. A sparse
regularizer based on Dirichlet distribution is incorporated and a
multiplicative algorithm for solving the corresponding optimization
problem is developed.

I find the introduction of Lemma 4 particularly interesting.
A direct use of the Dirichlet distribution with \alpha < 1 as a sparse
regularizer can lead the objective function to be non-convex and
unbounded.
This is obviously problematic when performing MAP estimation.
To avoid this, the authors has introduced a practically useful approach.
Under a slightly tighter constraint (X >= \eplison) than the
non-negativity constraint they successfully showed that Eqs 14-16 give
the global optimal solution to the corresponding constrained
optimization problem.
As the authors mention in the conclusion, it would be interesting to
see whether a similar multiplicative algorithm can be obtained with
other divergence measures than the generalized Kullback-Leibler
divergence. If this can be readily obtained, the Dirichlet sparsity
regularizer would become more practically useful in many situations.
Summary: This paper describes a new algorithm for the sparse multi-factor
nonnegative matrix factorization (mfNMF) problem, which generalizes
the original NMF problem to more than two factors. A sparse
regularizer based on Dirichlet distribution is incorporated and a
multiplicative algorithm for solving the corresponding optimization
problem is developed.
Author Feedback

Author rebuttal: We would like to thank all reviewers for the constructive comments. While we will correct typos and implement some suggestions in an improved version of this work, we provide responses to major concerns in the reviews here.

1 Both reviewer_5 and reviewer_6 questioned the motivation of using the MNIST data set in our experiment. We used the MNIST data set to visually demonstrate the effect of decomposing a nonnegative matrix using the proposed sparse multi-factor NMF algorithm. We thought these results, as images, may provide better intuitions to understand the effect of the algorithm, e.g., sparseness of the factors, though this experiment per se is not directly motivated by a particular application.

2 Reviewer_6 asked about the validity of Eq.(3). The extra term as pointed out by the reviewer vanishes due to the further decomposition we gave in the terms after the second equal sign of Eq.(2), (the proof is in A.2 of the supplementary materials).

3 Reviewer_6 asked about the relation mfNMF with two factors with other well known NMF algorithms with generalized KL objective. Existing NMF algorithms as we cited in the submission do not take advantage of the decomposition of the generalized KL divergence given in Eq(2), which reduces the problem to two sub-problems, each concerning only stochastic matrices or diagonal matrices. As such, our algorithm will lead to a different two factor NMF algorithm when compared to the existing methods.